# Roles of BrlA and AbaA in Mediating Asexual and Insect Pathogenic Lifecycles of *Metarhizium* *robertsii*

**DOI:** 10.3390/jof8101110

**Published:** 2022-10-21

**Authors:** Jin-Guan Zhang, Si-Yuan Xu, Sheng-Hua Ying, Ming-Guang Feng

**Affiliations:** Institute of Microbiology, Collegeof Life Sciences, Zhejiang University, Hangzhou 310058, China

**Keywords:** Hypocreales, insect pathogens, asexual developmental activators, gene expression and regulation, asexual cycle, infection cycle, virulence, stress tolerance

## Abstract

BrlA and AbaA are key activators of the central developmental pathway (CDP) that controls asexual development in *Aspergillus* but their roles remain insufficiently understood in hypocerealean insect pathogens. Here, regulatory roles of BrlA and AbaA orthologs in *Metarhizium* *robertsii* (Clavicipitaceae) were characterized for comparison to those elucidated previously in *Beauveria bassiana* (Cordycipitaceae) at phenotypic and transcriptomic levels. Time-course transcription profiles of *brlA*, *abaA*, and the other CDP activator gene *wetA* revealed that they were not so sequentially activated in *M.* *robertsii* as learned in *Aspergillus*. Aerial conidiation essential for fungal infection and dispersal, submerged blastospore production mimicking yeast-like budding proliferation in insect hemocoel, and insect pathogenicity via cuticular penetration were all abolished as a consequence of *brlA* or *abaA* disruption, which had little impact on normal hyphal growth. The disruptants were severely compromised in virulence via cuticle-bypassing infection (intrahemocoel injection) and differentially impaired in cellular tolerance to oxidative and cell wall-perturbing stresses. The Δ*brlA* and Δ*abaA* mutant shad 255 and 233 dysregulated genes (up/down ratios: 52:203 and 101:122) respectively, including 108 genes co-dysregulated. These counts were small compared with 1513 and 2869 dysregulated genes (up/down ratios: 707:806 and 1513:1356) identified in Δ*brlA* and Δ*abaA* mutants of *B. bassiana*. Results revealed not only conserved roles for BrlA and AbaA in asexual developmental control but also their indispensable roles in fungal adaptation to the insect-pathogenic lifecycle and host habitats. Intriguingly, BrlA- or AbaA-controlled gene expression networks are largely different between the two insect pathogens, in which similar phenotypes were compromised in the absence of either *brlA* or *abaA*.

## 1. Introduction

*Beauveria* and *Metarhizium* include species that represent classic insect pathogens of Cordycipitaceae and Clavicipitaceae (Hypocreales), respectively, and serve as main sources of fungal insecticides and acaricides crucial for green agriculture [1,2]. Aerial conidia of such fungi are usually produced as the active ingredient of fungal pesticides on artificial substrata, such as small grains [3]. Despite similar lifestyles, *Metarhizium* is considered to have evolutionarily acquired insect pathogenicity from plant-pathogenic or entophytic fungi approxiamtely130 million years (MY) later than *Beauveria* [2,4]. Perhaps due to this difference, the two fungal lineages have many sets of genes that are functionally differentiated or even differed in cellular processes and events vital for their adaptation to broad or specific host spectra and habitats [5,6,7,8]. They also feature different types of asexual developmental structures. *Metarhizium* spp. produce chained conidia on phialides, i.e., conidiophores developing from hyphae, while *Beauveria* spp. produce conidia on tiny zigzag rachises (conidiophores) and form spore balls (combination of conidia and zigzag rachises), which increase steadily in size and density during conidiation and are scattered when matured. However, little is known about any differences in genetic control of asexual development between the two lineages. Understanding such differences is important for improved production technology of high-quality conidia.

Aerial conidiation and conidial maturation is controlled by the central developmental pathway (CDP) in model *Aspergillus* and *Penicillium* species which form phialides and chained conidia (reviewed in refs. [9,10]). In *Aspergillus nidulans*, CDP is activated by BrlA to initiate conidiophore development [11], followed by sequential activation of AbaA and WetA in the middle and late phases of conidiophore development [12,13,14]. The sequential BrlA–AbaA–WetA activation leads to transcriptional activation of downstream genes involved in not only initiation of conidiation but also synthesis of spore wall components [15,16,17]. As a hallmark CDP player, BrlA is activated by three cascades consisting of FluG and FlbA–FlbE, which function in upstream developmental activation pathway (UDAP) [18,19,20,21,22,23,24,25,26]. While the genetic control principles on asexual development of *A. nidulans* are established in pioneering studies, little is known about the evolution of gene regulatory networks associated with the diversity of asexual developmental and morphological structures in filamentous fungi [27]. Indeed, it is increasingly concerned whether the principles elucidated in *A. nidulans* are applicable to Pezizomycotina due to increasing evidence of remarkable genome divergence in ascomycetes [27,28,29,30]. As one of three CDP activators, for example, AbaA is revealed to have been independently lost in four lineages of Eurotiomycetes, including the whole order Onygenales and those unable to form phialides in other orders [30].

Homologs of those CDP and UDAP activators well characterized in *A. nidulans* exist in both *Beauveria* and *Metarhizium* and have been studied in *Beauveria bassiana**,* which is a main source of wide-spectrum fungal pesticides. In *B. bassiana*, BrlA and AbaA act as master regulators of asexual developmental processes since aerial conidiation and submerged blastospore production essential for normal cuticle infection and subsequent hemocoel colonization were abolished in association with dysregulation of 1513 and 2869 genes in the knockout mutants of *brlA* and *abaA,* respectively [31]. WetA and downstream VosA also play essential roles in fungal conidiation and conidial maturation [32]. Despite similar regulatory roles for the CDP activators in asexual development of *B. bassiana* as seen in *A. nidulans*, the homologs of FluG and FlbA to FlbE have been shown to play no role in the activation of CDP, although they are required for fungal insect pathogenic lifestyle [33,34,35]. Nonetheless, only AbaA has been reported to regulate conidiation in *M. robertsii* and *M. acridum*, two insect pathogens previously classified to *Metarhizium* *anisopliae* sensu latu [36], and to bind the promoter region of *veA* in *M. robertsii* [37] or to be negatively mediated by NsdD in *M. acridum* [38]. However, other CDP activators remain functionally unexplored yet in *Metarhizium* spp. The different structures of *Beauveria* and *Metarhizium* in asexual development suggest possible differences in gene networks regulated by their key CDP activators. This study aimedto characterize functions of both BrlA and AbaA in *M. robertsii* and compare gene expression networks regulated by either activator with those shown previously in *B. bassiana* in order to reveal possible differences in genetic control of asexual development between the two insect pathogens.

## 2. Materials and Methods

### 2.1. Domain Analysis of brlA and abaA and Transcriptional Profiling of Their Genes

BrlA and AbaA orthologs were identified in the genome of *M. robertsii* [39] by BLASTp analysis (http://blast.ncbi.nlm.nih.gov/blast.cgi, accessed on 18 October 2022) using the amino acid sequences of *A. nidulans*BrlA (XP_658577) and AbaA (XP_658026) as queries. Conserved domains of both orthologs predicted at http://smart.embl-heidelberg.de/ (accessed on 18 October 2022) were compared with those of the queries and the counterparts characterized previously in *B. bassiana* [31]. A nuclear localization signal (NLS) motif in the sequence of each ortholog was predicted with a maximal probability at https://www.novopro.cn/tools/nls-signal-prediction (accessed on 18 October 2022) for comparison among the mentioned fungal species.

To reveal whether the three CDP genes *brlA*, *abaA*, and *wetA* are sequentially activated, the wild-type strain *M. robertsii* ARSEF 2575 (WT herein) was grown on the cellophane-overlaid plates of potato dextrose agar (PDA) by spreading 100 μL aliquots of a 10^7^ conidia/mL suspension per plate (*ϕ* = 9 cm) at the optimal regime of 25 °C and 12:12 (light:dark) photoperiod. During the period of a 7-day incubation, total RNA was extracted daily from each of three plate cultures using an RNAiso Plus Kit (TaKaRa, Dalian, China) and reversely transcribed into cDNA with a PrimeScript RT reagent Kit (TaKaRa). The transcripts of each target gene in three cDNA samples per day were quantified via real-time quantitative PCR (qPCR) with paired primers (Appendix A) under the action of SYBRPremix *Ex Taq* (TaKaRa) and normalized based on the fungal 18S rRNA. Relative transcript levels of analyzed genes over the period of incubation were assessed with respect to a standard level at the end of a 2-day incubation using the 2^−ΔΔCT^ method.

### 2.2. Subcellular Localization of brlA and abaA

The promoter P*tef1* of *tef1*, a gene-encodingtranslation elongation factor 1 alpha, was amplified from the WT DNA to take the place of the *B. bassiana* P*tef1* upstream of the C cassette (5′-EcoRI-XmaI-BamHI-PstI-HindIII-3′) in pAN52-C-gfp-bar [40]. Subsequently, the open reading frame of *brlA* or *abaA* amplified from the WT cDNA was ligated to the 5′-terminus of the green fluorescence protein gene *gfp* (GenBank U55763) at the enzyme sites of *Xma*I*/Bam*HI. The constricted vector pAN52-C-*x*-gfp-bar (*x* = *brlA* or *abaA*) was integrated into the WT strain using a method of transformation regulated by *Agrobacterium tumefaciens* [41]. The putative transgenic colonies of each transformation were screened by the *bar* resistance to phosphinothricin (200 μg/mL). A colony chosen with its strong green fluorescence was incubated for conidiation on PDA at the optimal regime. Conidia from the culture were suspended in Sabouraud dextrose both (4% glucose and 1% peptone) plus 1% yeast extract (SDBY) for a 3-day incubation on a shaking bed (150 rpm) at 25 °C. Hyphae taken from the liquid culture were rinsed repeatedly in dd-H_2_O and stained with DAPI (4′,6′-diamidine-2′-phenylindole dihydrochloride) of 4.16 mM, followed by visualization with laser scanning confocal microscopy (LSCM) at the excitation/emission wavelengths of 358/460 and 488/507 nm to determine subcellular localization of BlrA-GFP or AbaA-GFP.

### 2.3. Generation of brlA and abaA Mutants

The target gene *brlA* or *abaA* was disrupted from the WT strain by homologous recombination of the 5′ and 3′ coding/flanking fragments separated by the reporter gene *bar* in p0380- 5′*x*-bar-3′*x* (*x* = *brlA* or *abaA*), as illustrated in Appendix A. The deletion vector was constructed by amplifying the 5′ and 3′ fragments of each target gene from the WT DNA and inserting into appropriate enzyme sites (*Bam*HI*/Pst*Iand *Xho*I*/Xma*I) of linearized p0380-bar, and transformed into the WT conidia for homologous recombination. To complement *brlA* or *abaA* into its null mutant, a full-length coding sequence of each target gene with flank regions was amplified from the WT DNA and ligated into linearized p0380-sur-gateway to substitute the gateway fragment under the action of Gateway BP Clonase II Enzyme Mix (Invitrogen, Shanghai, China). The resultant vector p0380- sur-*x* was ectopically integrated into the protoplasts of an identified Δ*brlA* or Δ*abaA* mutant, which produced neither aerial conidia nor submerged blastosporers, by means of polyethylene glycol-mediated transformation [42]. The used protoplasts were prepared as described previously [31]. Putative deletion and complementation mutant colonies were screened respectively by the *bar* resistance to phosphinothricin (200 μg/mL) and the *sur* resistance to chlorimuron ethyl (10 μg/mL), identified via PCR analysis (Appendix A), and verified via qPCR analysis (Appendix A). All of the paired primers used for the manipulation and detection of *brlA* or *abaA* are listed in Appendix A. The identified Δ*brlA* and Δ*brlA::brlA* mutants and three Δ*abaA* mutants (Δ*abaA* 1–3), in which targeted gene complementation was not successful although many attempts were carried out, were used together with the WT strain in the following experiments. Each experiment includes three independent replicates per strain to meet a requirement for one-way analysis of variance and Tukey’s honestly significant difference (HSD) test.

### 2.4. Assays for Radial Growth Rates

For all tested strains, 100 μL aliquots of a hyphal suspension (fresh hyphal weight: 50 mg/mL; the same below unless specified otherwise) from a 3-day-old SDBY culture were spread onto cellophane-overlaid SDAY (i.e., SDBY plus 1.5% agar) plates and incubated at 25 °C for 3 days. Hyphal mass discs (5 mm diameter) taken from each culture with a cork borer were individually attached to the plate center of PDA, SDAY, 1/4 SDAY (amended with one-fourth of each SDAY nutrient), Czapek-Dox agar (CDA; 3% sucrose, 0.3% NaNO_3_, 0.1% K_2_HPO_4_, 0.05% KCl, 0.05% MgSO_4_, 0.001% FeSO_4_ and 1.5% agar) and CDAs amended with different amino acids as nitrogen sources. All inoculated plates were incubated for 7 days at the optimal regime, followed by estimation of each colony diameter with two measurements taken perpendicular to each other across the center.

The same method was used to initiate radial growth of each strain on CDA plates supplemented with H_2_O_2_ (2 mM), menadione (0.03 mM), Congo red (1 mg/mL), and calcofluor white (15 μg/mL), respectively. After a 7-day incubation at the optimal regime, the diameter of each colony was assessed as aforementioned. The diameters of stressed colonies (*d*_s_) and unstressed control colonies (*d*_c_) were used to compute relative growth inhibition percentage ((*d*_c_ − *d*_s_)/*d*_c_ × 100) of each strain under a given chemical stress.

### 2.5. Assays for Yields of Aerial Conidia and Submerged Blastospores

The 100 μL aliquots of a hyphal suspension per strain were evenly spread on PDA plates overlaid with cellophane film and incubated for 15 days at the optimal regime. Three plugs (5 mm diameter) were taken from each of the 3-, 7-, and 15-day-old cultures using the cork borer. Conidia in each plug were released into 1 mL of aqueous 0.05% Tween 80 via a 10 min supersonic vibration. Three samples taken from the resultant suspension were used to assess conidial concentration using a hemocytometer, and the concentration was converted to the number of conidia produced per unit area (cm^2^) of plate culture. Meanwhile, 7- and 15-day-old plate cultures were collected and dried for 6 h at 70 °C, followed by assessment of biomass level per plate. In addition, samples taken from the 5-day-old cultures of each strain were stained with the cell wall-specific dye calcofluor white and examined for conidiation status under a microscope.

To assess submerged blastospore yields, hyphal cells collected from the 3-day-old SDBY cultures of each strain were rinsed twice with sterile water and resuspended in 50 mL aliquots (fresh hyphal mass 1 mg/mL) of fresh SDBY in 150 mL flasks. Possible blastospores in each culture were removed by filtration through lens clearing tissues. All flasks were incubated for 3 days by shaking (150 rpm) at 25 °C. Three 50 μL samples were taken from each of three flasks per strain. Blastospore concentration (no. blastospores/mL) in each sample was assessed using a hemocytometer.

### 2.6. Assays for Hyphal Pathogenicity and Infection-Required Enzyme Activity

Since aerial conidiation and submerged blastospore production were both abolished in the absence of *brlA* or *abaA*, the 3-day-old SDBY cultures of all tested strains were prepared and filtered as aforementioned to remove possible blastospores of two control (WT and Δ*brlA::brlA*) strains. The collected hyphae of each strain were suspended in 0.05% Tween 80 and standardized to a concentration of fresh weight 100 mg/mL. Normal cuticle infection (NCI) was initiated by immersing three groups of 35–40 *Galleria* *mellonella* larvae (4th instar) in 40 mL aliquots of each strain’s hyphal suspension for 15 s. Alternatively, a microinjector was used to inject 5 μL of each strain’s hyphal suspension (fresh weight 10 mg/mL) into the hemocoel of each larva in each group for cuticle-bypassing infection (CBI). All treated groups were held at the optimal regime. The survival or mortality records of each group were monitored every 12 or 24 h until the records were stabilized. The time-mortality trend in each group was subjected to modeling analysis for estimation of median lethal time (LT_50_) as an index of fungal virulence.

Total activities of extracellular enzymes (ECEs, involved in proteolysis, chitinolysis, and lipolysis) and subtilisin-like Pr1 proteases collectively required for NCI via cuticular penetration [43,44] were assessed from liquid cultures of each strain as described previously [45]. Briefly, 50 mL aliquots of a fresh hyphal 1 mg/mL suspension in CDB (i.e., agar-free CDA) amended with the sole nitrogen source of 0.3% bovine serum albumin (BSA) as an enzyme inducer were incubated at 25 °C for 3 days by shaking (150 rpm). The supernatant collected from each culture was used as a crude extract to measure optical densities at 442 and 410 nm (OD_442_ and OD_410_) as indices of total ECEs and Pr1 activities, respectively. One unit of enzyme activity was defined as an enzyme amount required for an OD_442_ or OD_410_ increase by 0.01 after 1 h reaction of each extract relative to a blank control. Hyphal mass collected from each culture was dried overnight at 70 °C, followed by quantification of its biomass level.

A status of each strain’s hemocoel colonization vital for mycosis development and host mummification was examined as described previously [46]. Briefly, hemolymph samples were taken from surviving larvae 6 days post-NCI and 3 days post-CBI. Hemolymp of 100 μL from each strain-infected larvae was added to 3 mL SDBY containingkanamycin (100 ng/mL) and ampicillin (200 ng/mL). After a 36 h shaking incubation at 25 °C, all cells were collected from the culture and resuspended in 0.05% Tween 80. The status and abundance of fungal cells and insect hemocytes in the suspension were examined under a microscope.

### 2.7. Transcriptomic Analysis

Four-day-old cultures of the Δ*brlA*, Δ*abaA*-1, and WT strains (three cultures per strain) grown on cellophane-overlaid PDA plates at the optimal regime were sent to Lianchuan BioTech Company (Hangzhou, China) for construction and analysis of *brlA*- and *abaA*-specific transcriptomes as described previously [31,46]. Clean tags generated by filtration of all raw reads from sequencing were mapped to the *M. robertsii* genome [39]. Differentially expressed genes (DEGs) were identified at the significant levels of both log_2_ ratio (fold change) ≤−1 or ≥ 1 and *q* < 0.05, and then enriched to GO terms in three function categories (*p* < 0.05) via Gene Ontology (GO) analysis (http://www.geneontology.org, accessed on 18 October 2022) and pathways (*p* < 0.05) via Kyoto Encyclopedia of Genes and Genomes (KEGG) analysis (http://www.genome.jp/kegg, accessed on 18 October 2022).

## 3. Results

### 3.1. Domain Architecture, Transcription Profiles, and Subcellular Localization of brlA and abaA

The amino acid sequences of BrlA (KHO11655) and AbaA (EFZ03620) orthologs identified in the NCBI database of *M. robertsii* [39] were much less identical to the queries of *A. nidulans* (36.57% and 41.9%; e-values: 12 × 10^−09^ and 8 × 10^−12^; scores: 57.4 and 67.8; coverage: 20% and 10%, respectively) than those (72.46% and 68.37%; e-values: 7 × 10^−157^ and zero; scores: 444 and 1174; coverage: 100% and 100%) studied previously in *B. bassiana* [31]. Despite low sequence identity, the identified BrlA and AbaA orthologs (Figure 1A) featured two typical zinc-finger (ZnF-C2H2) domains at C-termini and an N-terminal TEA domain required for DNA binding [47], respectively. Notably, the BrlA zinc-finger domains in the three fungi were similar in size (23–26 amino acids (aa) per capita) while the AbaA TEA domain differed largely in molecular size between *A. nidulans* (70 aa) and the two insect pathogens (125 or 131 aa). Interestingly, a maximal probability predicted for an NLS motif of BrlA or AbaA in *B. bassiana* (0.87 or 0.92) or *M. robertsii* (0.91 or 0.89) was much greater than that in *A. nidulans* (0.44 or 0.32). For the *brlA* orthologs, additionally, the putative transcription units *brlAα*, *brlAβ*, and μORF found in *M. roberstii* showed low homology to those identified previously in *A. nidulans* [48] (Appendix A).

The *brlA* and *aba*A genes expressed in the WT strain showed differential time-course transcriptional profiles over the period of a 7-day incubation at the optimal regime (Figure 1B). Their transcript levels were not significantly upregualted until day 4 in comparison to the standard level on day 2, and the initial upregulation was one day later than that of *wetA*, the other CDP activator gene presumably downstream of *abaA*. Such transcription profiles demonstrated earlier activation of *wetA* than the simultaneous activation of *brlA* and *abaA*, implying that the three CDP genes were not sequentially activated in *M. roberstii* as elucidated in *A. nidulans* [9,10]. Instead, they seemed to be activated in a fashion independent of one another.

A nuclear localization of either BrlA or AbaA suggested by its NLS motif predicted at a high probability was confirmed by its GFP-tagged fusion protein expressed in the WT strain. Both BrlA-GFP and AbaA-GFP (shown in green) accumulated more in the nuclei than in the cytoplasm of hyphal cells stained with the nuclear dye DAPI (shown in red), forming a distinctively merged yellow color in the nuclei (Figure 1C). The confirmed nuclear localization suggested a role of BrlA or AbaA in certain nuclear events including mediation of gene expression.

### 3.2. Distinctive Roles of brlA and abaA in Radial Growth and Stress Response

The Δ*brlA* and three Δ*abaA* mutants and the control (WT and Δ*brlA::brlA*) strains showed similar growth rates and colony morphology during a 7-day incubation under normal culture conditions after their growths were initiated by hyphal mass discs attached to the plates of rich and minimal media (Figure 2A). The diameters of their 7-day-old colonies formed on each of the media PDA, SDAY, 1/4 SDAY, CDA, and CDAs amended with different amino acids as nitrogen sources (Figure 2B−D) were not significantly different from one another (*p* > 0.05 in Tukey’s HSD tests). Compared with the control strains, however, the Δ*brlA* mutant was significantly more sensitive to oxidative stress induced by H_2_O_2_ or menadione (*p* < 0.05 in Tukey’s HSD tests) and much more sensitive to cell wall stress induced by Congo red or calcofluor white (*p* < 0.001 in Tukey’s HSD tests) (Figure 2E). In contrast, three Δ*abaA* mutants exhibited a similar increase only in sensitivity to calcofluor white among the tested chemical stressors on CDA plates.

The presented data revealed the greater role of *brlA* in the fungal response to cell wall perturbing stress than to oxidative stress, significant role of *abaA* in response to cell wall stress of calcofluor white, but dispensable role of either *brlA* or *abaA* in sustaining the fungal growth under normal culture conditions.

### 3.3. Indispensable Roles of brlA and abaA in Asexual Spore Production

The aerial conidiation level of each tested strain was assessed during a 15-day incubation on PDA at the optimal regime after initiation of its cultures by spreading 100 μL aliquots of a uniform hyphal suspension. Conidiophores and conidia were present in the samples taken from the 5-day-old cultures of the control strains and stained with a cell wall-specific dye but not in the Δ*brlA* and Δ*abaA* mutants’ samples, in which thicker hyphae were not differentiated at all (Figure 3A). The 15-day-old cultures showed some differences in morphology among the tested strains (Figure 3B). The control strains’ conidiation statuses were well shown in their darkening cultures. In contrast, the Δ*brlA* culture was consistently whitish. The Δ*abaA* mutants’ cultures were also whitish but yellowish in part, suggesting increased production of certain pigments in the absence of *abaA*. The biomass levels measured from the 7- and 15-day-old cultures of the Δ*abaA* mutants were significantly higher (*p* < 0.05 in Tukey’s HSD tests) than those not significantly different (*p* > 0.05) between the corresponding cultures of Δ*brlA* and its control strains (Figure 3C). The mean (±SD) conidial yield produced by the control strains was 1.52 (±0.53) × 10^5^ conidia/cm^2^ (*n* = 6) in their 3-day-old cultures and increased to 1.86 (±0.21) × 10^7^ and 2.88 (±0.27) × 10^7^ conidia/cm^2^ in their 7- and 15-day-old cultures respectively (Figure 3D). In contrast, aerial conidiation was completely abolished in Δ*brlA* and Δ*abaA* mutants’ cultures during the 15-day incubation.

Submerged blastospore production indicates an ability for fungal cells to proliferate rapidly by yeast-like budding for host mummification to death after hyphal invasion into the host body and is controlled by BrlA and AbaA in *B. bassiana* [31]. In the present study, the mean blastospore concentration measured from the 3-day-old SDBY cultures of the control strains was 6.83 (±0.15) × 10^3^ blastospores/mL (*n* = 6) while submerged blastospore production mimicking fungal proliferation in insect hemolymph was abolished in the Δ*brlA* and Δ*abaA* mutants’ cultures (Figure 3E).

Altogether, BrlA and AbaA were indispensable for aerial conidiation and submerged blastospore production of *M. roberstii* as elucidated previously in *B. bassiana* [31]. In other words, BrlA and AbaA serve as regulators of asexual developmental processes.

### 3.4. Indispensable Roles of brlA and abaA in Insect-Pathogenic Lifecycle

In the bioassays with a standardized hyphal suspension, the control strains killed all *G.* *mellonella* larvae within 18 days post-NCI or 12 days post-CBI (Figure 4A), resulting in a mean LT_50_ (*n* = 6) of 6.99 (±0.89) days via NCI and 3.38 (±0.15) via CBI (Figure 4B). In contrast, few tested larvae died from the Δ*brlA* and Δ*abaA* mutants 20 days post-NCI, and the mean surviving percentages were up to 63% (±6.3, *n* = 3) and 59% (±6.0, *n* = 9) for the Δ*brlA* and Δ*abaA* mutants 13 days post-CBI, respectively. Therefore, no LT_50_ was accessible for any of the mutants against the model insect via NCI or CBI.

For insight into a complete loss of the mutant’s pathogenicity via NCI, total activities of NCI-required ECEs and Pr1 proteases were assessed from the supernatants of the 3-day-old cultures initiated with a standardized hyphal suspension in CDB-BSA. Surprisingly, no significant variability was found in the activities of either ECEs (*F*_5,12_ = 0.79, *p* = 0.58) or Pr1 proteases (*F*_5,12_ = 1.46, *p* = 0.27) among all of the mutant and control strains tested (Figure 4C). Neither was a significant variability found among the biomass levels (*F*_5,12_ = 1.14, *p* = 0.39) in their CDB-BSA cultures.

Next, a status of fungal hemocoel colonization was observed in the hemolymph samples taken from the larvae surviving 6 days post-NCI or 3 days post-CBI. After a 36 h shaking incubation of the samples in SDBY at 25 °C, the samples contained observable yeast-like budding cells formed by the control strains (Figure 4D). In contrast, the Δ*brlA* and Δ*abaA* mutants’ budding cells were not observable in the samples examined post-NCI and very few in the samples examined post-CBI.

All of these observations indicated an indispensability of either BrlA or AbaA for the insect-pathogenic lifecycle of *M. robertsii*. The marked attenuation of the Δ*brlA* and Δ*abaA* mutants’ virulence via CBI was attributable to a blockage of their proliferation in host hemoceol. However, the mutants’ ECEs and Pr1 activities not significantly different from the control strains’ counterparts revealed little clue to a complete loss of the mutants’ pathogenicity to the model insect via NCI. Other NCI-related cellular events, including conidial adhesion to, germination on, and hyphal invasion into insect integument, were hardly observable when asexual spore production was abolished.

### 3.5. Transcriptomic Insight into Regulatory Roles of brlA and abaA in M. robertsii

There were 255 and 233 DEGs (up/down ratios: 52:203 and 101:122 respectively; the same meaning for ratios mentioned below) identified from the transcriptomes of the Δ*brlA* and Δ*abaA*-1 mutants versus the WT strain, respectively (Figure 5A; Appendix A). Intriguingly, 108 DEGs appeared in both Δ*brlA* (19:89) and Δ*abaA*-1 (19:89), including 28 annotated as hypothetic proteins or functionally unknown; most of them were co- upregulated or co-downregulated in the two mutants (Appendix A). The counts of DEGs were small in comparison to 1513 (707:806) and 2869 (1513:1356) DEGs identified in the previous Δ*brlA* and Δ*abaA* transcriptomes of *B. bassiana* [31]. The counts of DEGs and the different up/down ratios in the present and previous Δ*brlA* and Δ*abaA* mutants suggest that gene expression networks controlled by either BrlA or AbaA differ largely between *M. robertsii* and *B. bassiana*, although both of them are hypocrealean insect pathogens.

Previously, transcriptomic analysis revealed differential repression of other CDP activator genes in the absence of *brlA* or *abaA* in *B. bassiana* [31]. In the present study, surprisingly, the other CDP genes were not present or downregulated in either Δ*brlA* or Δ*abaA*. This reinforced a likelihood that they could be activated independently as revealed by their time-course transcription profiles in the WT strain.

The GO analysis resulted in 34 and 10 terms enriched to three GO categories of Δ*brlA* (Appendix A) and Δ*abaA*-1 (Appendix A) at the significance of *p* < 0.05, respectively. The Δ*brlA* mutant had 138 (26:112), 114 (34:84) and 119 (29:90) DEGs enriched to 3, 15, and 16 terms of cellular component, biological process, and molecular function, respectively (Figure 5B). The terms of cellular component included cellular component (23:94), integral component of membrane (3:11), and plasma membrane (0:7). The main terms of biological process were biological process (16:52), obsolete oxidation-reduction process (4:10), cellular response to xenobiotic stimulus (0:5), and emericellamide biosynthetic process (0:3); many other terms in the category contained only one or two DEGs. The main terms of molecular function were molecular function (17:57), nucleotide binding (2:4), ATP hydrolysis activity (0:4), ABC-type xenobiotic transporter activity (0:4), nucleoside–triphosphatase activity (1:3) and iron ion binding (2:3). The very low up/down ratios in the mentioned GO terms implicated that cell component, biological process and molecular function were severely compromised in the absence of *brlA*. Despite fewer DEGs (76:104) enriched to limited GO terms, the Δ*abaA* mutant had three cellular component terms compromised (Figure 5C), including cellular component (39:46), integral component of membrane (3:7), and cytoplasmic vesicle (1:1). In Δ*abaA*-1 mutant, only two terms of biological process, i.e., emericellamide biosynthetic process (2:1) and obsolete pathogenesis (3:3), were affected while five molecular function terms were compromised, including mainly molecular function (23:40), acyltransferase activity (2:2) and glutathione transferase activity (0:2). The GO analysis revealed more profound impact of *brlA* disruption on gene expression networks than of *abaA* disruption in *M. robertsii* despite overlapping effects on the terms of cellular component, integral component of membrane, and molecular function. Indeed, the mentioned main GO terms comprised most of those genes co-dysregulated in both Δ*brlA* and Δ*abaA*-1.

Unexpectedly, the KEGG analysis revealed very limited information due to only a few DEGs enriched to two or three pathways at the significance of *p* < 0.05 (Figure 5D). The pathway of ABC transporters (map02010) was co-repressed in Δ*brlA* (1:5) and Δ*abaA*-1 (1:3). One or two other pathways enriched were ascorbate and aldarate metabolism (map00053, 2:1) and fatty acid biosynthesis (map00061, 0:3) in Δ*brlA*, and staurosporine biosynthesis (map00404, 3:1). The co-repressed pathway was seemingly related to the malfunction of the GO term known as an integral component of the membrane.

## 4. Discussion

In the present study, BrlA and AbaA were shown to localize in both the nuclei and the cytoplasm of hyphal cells and proved indispensable for the asexual cycle in vitro and in vivo but nonessential for hyphal growth in *M. robertsii* as characterized previously in *B. bassiana* [31]. The indispensability is highlighted by the Δ*brlA* and Δ*abaA* mutants’ inability to produce aerial conidia and submerged blastospores and to invade into insect body via cuticular penetration as well as their impaired capability of host hemocoel colonization. The Δ*brlA* mutant was significantly compromised in cellular tolerance to two oxidants and two cell-wall-perturbing agents, contrasting to an increased sensitivity of the Δ*abaA* mutants to only a calcofluor white-induced stress. The present and previous studies reinforce not only conserved roles of BrlA and AbaA in regulating asexual development in hypocrealean insect pathogens as seen in model fungi [9,10] but also their essential roles in fungal adaptation to insect-pathogenic lifestyle and the environment. While similar phenotypes appeared in Δ*brlA* and Δ*abaA* mutants, gene expression networks controlled by either BrlA or AbaA differed largely between the two insect pathogens, as discussed below.

First, the expression of *brlA* and *abaA* in *M. robertsii* was not upregulated until the end of a 4-day incubation and the upregulation was one day later than that of *wetA*, which was also activated earlier than *abaA* in *B. bassiana* [32]. Such time-course transcription profiles suggest non-sequential activation of the three CDP genes in *M. robertsii* and also the reason for using 4-day-old cultures for transcriptomic analysis in this study. Indeed, AbaA has been shown to bind the promoter region of *veA* in *M. robertsii* [37] and to be negatively mediated by NsdDin *M. acridum* [38]. The UDAP regulators FluG and FlbA to FlbE are known to activate the expression of *brlA* responsible for sequential activation of *abaA* and *wetA* in *A. nidulans* [9,10] but have proved unable to do so in *B. bassiana* [33,34,35]. The activation of *brlA* for initiation of asexual development in *B. bassiana* seems to be achieved via multiple routes or pathways other than UDAP, as discussed previously [31,33,34,35]. The previous and present studies suggest an infeasibility for sequential activation of three CDP genes in *B. bassiana* and *M. robertsii*.

Moreover, gene expression networks controlled by BrlA and AbaA are greatly simplified in *M. roberstii* in comparison to those in *B. bassiana* [31]. This is presented by much smaller counts of DEGs in the Δ*brlA* and Δ*abaA* mutants of *M. robertsii* than of *B. bassiana*. Interestingly, enriched GO terms inΔ*brlA* (34) were far greater than those in Δ*abaA* (5) in *M. roberstii*. However, the situation was reversed in *B. bassiana*, namely 5 and 29 GO terms enriched toΔ*brlA* and Δ*abaA* respectively [31]. Most of those GO terms enriched to either Δ*brlA* or Δ*abaA* were different between *M. robertsii* and *B. bassiana* irrespective of being categorized to cellular component, biological process, or molecular function. In a previous study to characterize the key transcription factor Msn2 downstream of the MAPK Hog1 signaling cascade, the counts of identified DEGs were 3% smaller and 12% greater in the Δ*msn2* mutant’s responses of *M. robertsii* than of *B. bassiana* to oxidative stress and heat shock, respectively [49]. In addition, up to 1818 DEGs (1006:801) were identified from the deletion mutant’s transcriptome of *cfp*, a gene encoding small cysteine-free protein confirmed as a virulence factor in *B. bassiana* [50], while only 604 DEGs (251:353) were found in the deletion mutant’s transcriptome of a *cfp* homolog also acting as a virulence factor in *M. robertsii* [46]. In the present and previous studies, gene expression networks either controlled by BrlA and AbaA for asexual development and those controlled by Msn2 for stress responses or by CFP required for virulence are largely different between *M. robertsii* and *B. bassiana*. Such differences are likely due to a 130-MY difference of evolution histories in their adaptation to the insect-pathogenic lifestyle and host habitats [2,4].

Furthermore, the majority of genes dysregulated in the Δ*brlA* and Δ*abaA* mutants of *M. robertsii* were enriched to only a few main GO terms, including cellular component, integral component of membrane, molecular function, and biological process collectively responsible for their phenotypic changes. Most of those co-dysregulated genes appeared in the main GO terms of the two mutants. Apart from those encoding hypothetic or unknown proteins, very limited DEGs were found to involve in specific cellular processes and events associated with abolished pathogenicity and blocked hemocoel colonization of each mutant. This is also different from marked linkages of hundreds of dysregulated genes to main phenotypic defects in the same mutants of *B. bassiana* [31].

Conclusively, BrlA and AbaA serve as master regulators of asexual development and insect pathogenic lifecycle in *M. robertsii* as they do in *B. bassiana*. However, gene expression networks controlled by BrlA or AbaA in *M. robertsii* are much simplified in comparison to those controlled by either ortholog in *B. bassiana* that could have adapted to the insect-pathogenic lifestyle ~130 MY earlier [2,4]. This finding sheds light on substantial differences of the key CDP activators-governed genetic backgrounds between the two insect pathogens as representatives of Cordycipitaceae and Claviciptaceae in Hypocreales.

## Figures and Tables

**Figure 1 jof-08-01110-f001:**
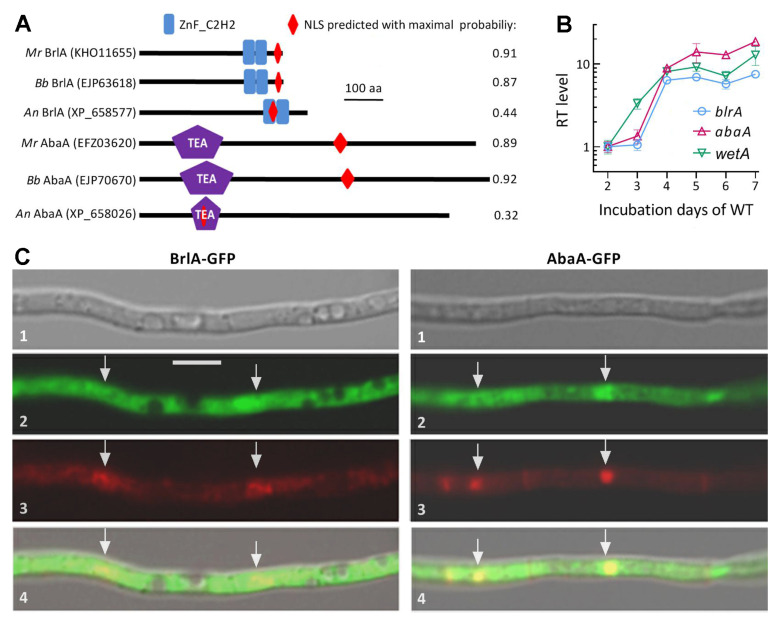
Domain architectures, transcriptional profiles, and subcellular localization of BrlA and AbaA in *M. robertsii* (*Mr*). (**A**) Comparison of conserved domains and NLS motif predicted from amino acid sequences of BrlA and AbaA orthologs. *An*, *A. nidulans*. *Bb*, *B. bassiana*. (**B**) Relative transcript (RT) levels of three CDP genes in the *Mr* WT strain during a 7-day incubation on PDA at an optimal regime with respect to a standard on day 2. Error bars: standard deviations (SDs) of the means from three independent cDNA samples analyzed via qPCR. (**C**) LSCM images (scale bar: 5 μm) for subcellular localization of the BrlA-GFP and AbaA-GFP fusion proteins expressed in the WT strain. Images 1, 2, 3, and 4 are bright, expressed, DAPI-stained, and merged views of the same field, respectively. Hyphal nuclei are indicated by arrows.

**Figure 2 jof-08-01110-f002:**
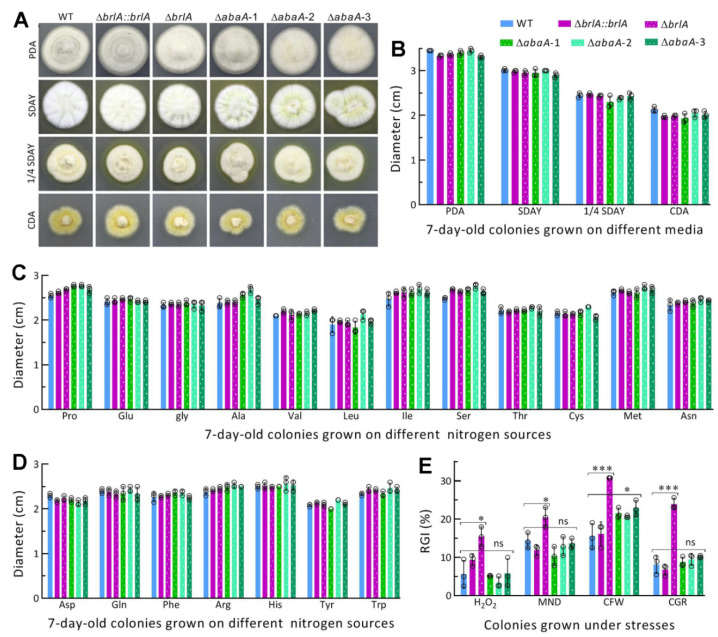
Impact of *brlA* or *abaA* disruption on radial growth and stress response of *M. robertsii*. (**A**,**B**) Images and diameters of fungal colonies grown at the optimal regime of 25 °C and L:D 12:12 for 7 days on nutrition-rich (PDA, SDAY, and 1/4 SDAY) and minimal (CDA) media. (**C**,**D**) Diameters of 7-day-old fungal colonies grown at the optimal regime on CDA amended with different amino acids as nitrogen sources. (**E**) Relative growth inhibition (RGI) percentages of fungal colonies grown at 25 °C for 7 days on CDA plates supplemented with H_2_O_2_ (2 mM), menadione (MND, 0.03 mM), calcofluor white (CFW, 15 μg/mL), and Congo red (CGR, 1 mg/mL) respectively. All colonies were initiated with hyphal mass discs (*ϕ* = 5 mm) attached to plates. *p* < 0.05 * or 0.001 *** in Tukey’s HSD tests (ns, no significance). Error bars: SDs from three independent replicates.

**Figure 3 jof-08-01110-f003:**
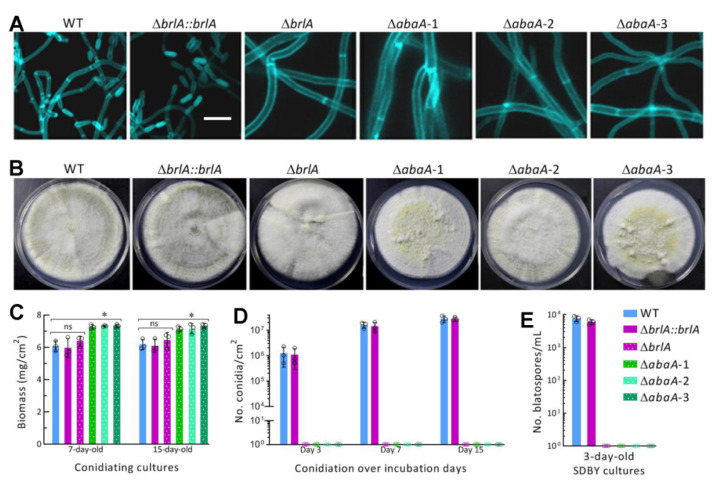
Indispensability of either *brlA* or *abaA* for asexual spore production of *M. robertsii*. (**A**,**B**) Microscopic images (scale bar: 10 μm) of culture samples stained with calcofluor white after collected from 5-day-old cultures and images of 15-day-old cultures grown on PDA at the optimal regime of 25 °C and L:D 12:12. (**C**,**D**) Biomass levels and conidial yields of fungal cultures assessed during a 15-day incubation on PDA at the optimal regime. Each culture was initiated by spreading 100 μL of a fresh hyphal 50 mg/mL suspension. (**E**) Blastospore yields measured from the 3-day-old SDBY cultures initiated with fresh hyphal mass 1 mg/mL. * *p* < 0.05 in Tukey’s HSD tests. Error bars: SDs from three replicates.

**Figure 4 jof-08-01110-f004:**
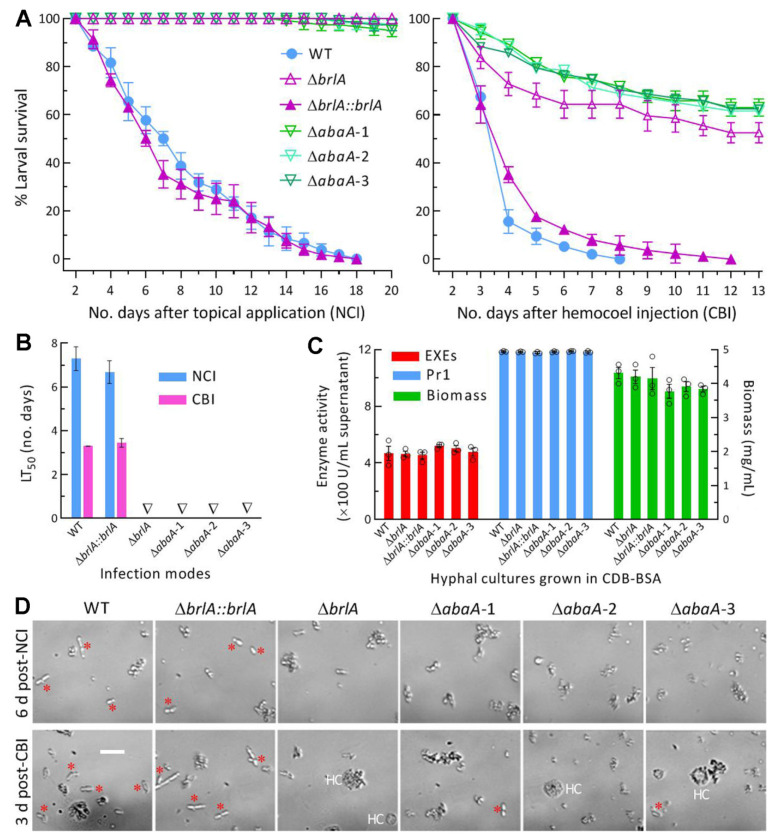
Indispensability of *brlA* or *AbaA* for insect-pathogenic lifecycle of *M. robertsii*. (**A**) Survival trends of *G. mellonella* larvaeafter immersion in a fresh hyphal 100 mg/mL suspension for normal cuticle infection (NCI) and intrahemocoel injection of 5 μL fresh hyphal (10 mg/mL) suspension per larva for cuticle-bypassing infection (CBI). (**B**) LT_50_ values estimated from time-mortality trends. Triangles indicate an LT_50_ not accessible for Δ*brlA* and Δ*abaA* mutants. (**C**) Total activities of cuticle degrading enzymes (ECEs and Pr1 proteases) and biomass levels assessed from the 3-day-old submerged cultures generated by shaking incubation of a fresh hyphal 1 mg/mL suspension in CDB-BSA. (**D**) Microscopic images (scale bar: 20 μm) for status and abundance of fungal budding cells (marked with red stars) and insect hemocytes (HC) in hemolymph samples, which were taken from surviving larvae 6 days post-NCI and 3 days post-CBI and incubated at 25 °C for 36 h in SDBY. Error bars: SDs from three independent replicates.

**Figure 5 jof-08-01110-f005:**
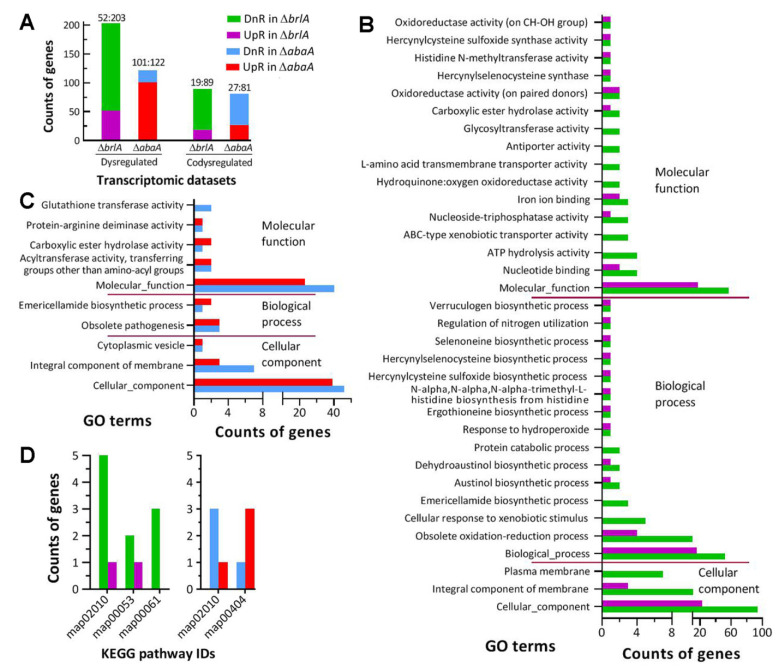
Effect of *brlA* or *abaA* disruption on gene expression networks of *M. robertsii*. (**A**) Counts of genes dysregulated and co-dysregulated in the Δ*brlA* and Δ*abaA*-1 mutants versus the WT strain. (**B**,**C**) Counts of dysregulated genes significantly enriched (*p* < 0.05) to GO terms of three function classes in Δ*brlA* and Δ*abaA*-1, respectively. (**D**) Counts of dysregulaated genes significantly enriched (*p* < 0.05) to KEGG pathways in the two mutants. The transcriptome was constructed based on three 4-day-old cultures (replicates) of the Δ*brlA*, Δ*abaA*-1,and WT strains. All dysregulated genes were identified at the significant levels of log_2_ ratio (fold change) ≤−1 (downregulated, DnR) or ≥1 (upregulated, UpR) and *q* < 0.05.

## Data Availability

All experimental data are included in this paper and Appendix A. All RNA-seq dataare available at the NCBI’s Gene Expression Omnibus under the accession PRJNA875283 (http://www.ncbi.nlm.nih.gov/bioproject/875283) aside from those reported in Appendix A of this paper.

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
