# Peer review of "Roles of BrlA and AbaA in Mediating Asexual and Insect Pathogenic Lifecycles of Metarhizium robertsii"

_jof, 2022, doi:10.3390/jof8101110_

Round 1

Reviewer 1 Report (Previous Reviewer 1)

In this study Zhang et al. identified the orthologs of Aspergillus brlA and abaA in Metarhizium robertsii, the major regulators of the asexual reproduction. The authors examined the transcription of that genes in different developmental stages and characterized the knock-out mutants, including conidiation and blastospore production, virulence, stress tolerance and transcriptomic profiles.

The manuscript is well-written, the figures are informative and while M. robertsii is used as a biocontrol agent against agricultural pests, deeper insight into the regulation of asexual reproduction is of great interest.

Major points:

Figure 1: subcellular localization of brlA and abaA was investigated with GFP fusion proteins. The quality of images (resolution) needs to be increased. On the other hand, on LSCM images fluorescence signal exist in the complete hyphae. Presumably, the proteins are localized in the nuclei thus the green color resulted by autofluorescence, which needs to be reduced. Moreover, colocalization on Figure1/4 is resulted by the intensive red signal of DAPI and not the GFP signal. The LSCM parameters (e.g., channels) must be also included in the Mat Methods.

Minor points/questions:

-A regulatory network figure would be useful in the part Introduction including the major regulators of conidiation in Aspergillus (including CDP, UDAP and downstream genes).

-In Aspergillus nidulans brlA consist of overlapping transcription units (brlAalfa, brlAbeta and a uORF located in the 5`UTR). Similar alternative transcripts of brlA also exist in Metarhizium? (if yes than it should be described in section 3.1.)

-When activation of CDP genes was examined L:D 12:12 photoperiod incubation condition was used. RNA was isolated during the light or the dark period?

-It is not clear from the text why complementation of abaA in the deletion mutant was unsuccessful. Short explanation needed here.

-Text needs to be revised; few typos exist.

Author Response

Response to Reviewer 1

Comments and Suggestions for Authors

In this study Zhang et al. identified the orthologs of Aspergillus brlA and abaA in Metarhizium robertsii, the major regulators of the asexual reproduction. The authors examined the transcription of that genes in different developmental stages and characterized the knock-out mutants, including conidiation and blastospore production, virulence, stress tolerance and transcriptomic profiles.

The manuscript is well-written, the figures are informative and while M. robertsii is used as a biocontrol agent against agricultural pests, deeper insight into the regulation of asexual reproduction is of great interest.

Author response: Thank you very much for your understanding and encouragement.

Major points:

Figure 1: subcellular localization of brlA and abaA was investigated with GFP fusion proteins. The quality of images (resolution) needs to be increased. On the other hand, on LSCM images fluorescence signal exist in the complete hyphae. Presumably, the proteins are localized in the nuclei thus the green color resulted by autofluorescence, which needs to be reduced. Moreover, colocalization on Figure1/4 is resulted by the intensive red signal of DAPI and not the GFP signal. The LSCM parameters (e.g., channels) must be also included in the Mat Methods.

Author response: The LSCM parameters used in subcellular localization have been added as suggested. The DAPI concentration used for nuclear staining is also added. I have tried to improve the quality of those LSCM images under the help of an LSCM expert but gained little progress in reducing some autofluorescence. Compared with our previous LSCM images for subcellular localization of the same fusion proteins in B. bassiana, a main conclusion on more accumulation of either BrlA-GFP or AbaA-GFP in the nuclei than the cytoplasm of M. robertsii is still supported by the LSCM images in Figure 1 although there is some autofluorescence hardly reduced.

Minor points/questions:

-A regulatory network figure would be useful in the part Introduction including the major regulators of conidiation in Aspergillus (including CDP, UDAP and downstream genes).

Author response: The regulatory network of CDP, UDAP and downstream genes in Aspergillus has been shown in excellent reviews, such as Etxebeste et al. (2010, 2019) and Park and Yu (2012) which we have cited in the Introduction section.

-In Aspergillus nidulans brlA consist of overlapping transcription units (brlAalfa, brlAbeta and uORF located in the 5`UTR). Similar alternative transcripts of brlA also exist in Metarhizium? (if yes than it should be described in section 3.1.)

Author response: We performed sequence alignment analysis to reveal whether brlA in M. robertsii vectors the mentioned transcription units brlAa, brlAb and mORF of its orthologue identified previously in A. nidulans (Prade and Timberlake1, 1993). We found low sequence identity in each unit between the two fungi, as shown in added Figure S2.

-When activation of CDP genes was examined L:D 12:12 photoperiod incubation condition was used. RNA was isolated during the light or the dark period?

Author response: RNA was extracted in the morning approximately 3 h after 12-h dark incubation.

-It is not clear from the text why complementation of abaA in the deletion mutant was unsuccessful. Short explanation needed here.

Author response: Targeted gene complementation was occasionally unsuccessful in our studies on hundreds of genes in fungal insect pathogens. In the present study, we tried all possible strategies in many attempts to complement abaA into its disruption mutant during a period of 15 months but always failed with no reason to be understood. In this case, we had to use three disruption mutants of abaA as replicates in the present study as we did occasionally in the previous studies.

-Text needs to be revised; few typos exist.

Author response: We have tried our best to revise the text and correct typos.

Reviewer 2 Report (New Reviewer)

Dear Drs

Happy day

The paper can be published as it is. Only few spelling mistakes should be correct.

And the contrast of the images need some adjustment.

Kindly add the alignment with actual amino acids sequences in the sublimity data

With my pleasure

Amro Amara    

Author Response

Comments and Suggestions for Authors

Dear Drs

Happy day

The paper can be published as it is. Only few spelling mistakes should be correct.

And the contrast of the images need some adjustment.

Kindly add the alignment with actual amino acids sequences in the sublimity data

With my pleasure

Amro Amara

Author response: Thank you very much for your understanding and encouragement. Your suggestions have been incorporated into this revision.

Reviewer 3 Report (New Reviewer)

The manuscript reports on the analysis of the functions of two core components controlling asexual reproduction in fungi, i.e. AbaA and BrlA, in the insect pathogenic fungus Metarrhizium robertsii.  The proteins were localized within the hyphal compartments, then the genes disrupted and phenotypes explored, showing a loss of the synthesis of asexual spores and in insect pathogenicity.  Last, the authors examined transcriptome of the strains to identify genes up or down regulated in response to the loss of the two genes.  There is considerable interest in spore development in fungi, so this work will be of interest to a large proportion of the fungal community.

Points for consideration are:

(1)    While one sees it on a regular basis in papers where people try to compare gene expression profiles across species, this is almost impossible to do without substantial additional experimentation.  First, both transcriptome sets have to be off material cultures side-by-side in the same growth conditions.  Second, additional time courses are needed in case there are differences in growth rates.  Then there are numerous other experiments that are required.  Furthermore, a stronger RNA-seq approach is to have the genes under control of a tightly regulated promoter, switch them on then look at short time points for gene upregulation as a better assessment of what genes are regulated.  Last, a powerful control is the gene switch, i.e. B. bassiana copy into the corresponding M. robertsii mutant (and vice versa) and examine phenotypes.  Hence, I strongly recommend rewriting lines 376-380, 482-487 and 498-504.

(2)    Line 24: ‘Intriguingly’ is perhaps a stretch, as it is well established that the CDP has different roles between fungal species, perhaps best illustrated in three species in the same genus (rather than two species in different families as here) by Wu et al. 2018 mBio doi.org/10.1128/mBio.01130-18.  This paper would be well worth considering in the Discussion.

(3)    It would be good to expand on the reason no AbaA complementation strains could be generated.

(4)    There are numerous typographic or other edits that would improve the manuscript, some of which are listed below.

General ‘UDAP’: is this a standard abbreviation in the field?  It becomes cumbersome on lines 55-57 or 66 where the ‘A’ duplicates ‘activates’.

Lines 61-62: could be ‘increasingly apparent that the principles elucidated in A. nidulans are applicable across the Pezizomycotina’.  Note this is a bit strange given the research performed on other models, like Neurospora crassa.  Worth considering other wording.

Line 67: ‘bassiana which is a main source’.

Line 76: ‘but are required for the’.

Line 78: ‘sensu lado’? likely delete.

Line 97: probably should write as ‘To reveal whether the transcription of three…’ then ‘are sequentially regulated’.

Line 111: may need italics for ‘Ptef1’.

Line 121: spelling ‘Sabouraud’.

Line 167: add space ‘(Ds) and’.

Line 196: delete ‘respectively’.

Line 246: could be ‘The brlA and abaA genes expressed in the wt strain showed differential time’.

Figure 2: the ‘A’ may need to be moved up to avoid overlap with the photograph.

Line 479: spelling ‘transcriptome’.

Lines 480 and 482: clear text by deleting ‘extraordinary’ and ‘vital’.

Line 489: ‘as they do in’.

Line 623: spelling ‘cysteine’.

Table S1 text: delete extra .  ‘plasmid..’.

Author Response

Response to Reviewer 3

Comments and Suggestions for Authors

The manuscript reports on the analysis of the functions of two core components controlling asexual reproduction in fungi, i.e. AbaA and BrlA, in the insect pathogenic fungus Metarrhizium robertsii. The proteins were localized within the hyphal compartments, then the genes disrupted and phenotypes explored, showing a loss of the synthesis of asexual spores and in insect pathogenicity.  Last, the authors examined transcriptome of the strains to identify genes up or down regulated in response to the loss of the two genes.  There is considerable interest in spore development in fungi, so this work will be of interest to a large proportion of the fungal community.

Author response: Thank you very much for your understanding and encouragement.

Points for consideration are:

(1    While one sees it on a regular basis in papers where people try to compare gene expression profiles across species, this is almost impossible to do without substantial additional experimentation.  First, both transcriptome sets have to be off material cultures side-by-side in the same growth conditions.  Second, additional time courses are needed in case there are differences in growth rates.  Then there are numerous other experiments that are required.  Furthermore, a stronger RNA-seq approach is to have the genes under control of a tightly regulated promoter, switch them on then look at short time points for gene upregulation as a better assessment of what genes are regulated.  Last, a powerful control is the gene switch, i.e. B. bassiana copy into the corresponding M. robertsii mutant (and vice versa) and examine phenotypes.  Hence, I strongly recommend rewriting lines 376-380, 482-487 and 498-504.

Author response: Good points! We agree with that there are possible effects of culture conditions and fungal growth rates on genome-wide gene expression. Therefore, our present and previous transcriptomic analyses were based on fungal cultures grown under optimal conditions. The 4-day-old cultures were chosen for transcriptomic analysis in this study since brlA and abaA were not largely upregulated in M. robertsii until the end of a 4-day incubation under optimal conditions (Figure 1B). The previous transcriptomes of B. bassiana brlA and abaA mutants were based on 84-h-old cultures grown under the same conditions.

The mentioned lines 376-380 and 498-504 in the PDF version we submitted are related to the legends of Figure 4 and the information of Supplementary Material, respectively, while lines 482-487 are associated with a discussion on phonotype-related genes in the brlA and abaA mutants of M. robertsii and B. bassiana. I guess that you were suggesting us to rewrite the sentences for phonotypic and transcriptomic comparisons between the two fungi. We have tried our best to do so in Discusion.

 (2) Line 24: ‘Intriguingly’ is perhaps a stretch, as it is well established that the CDP has different roles between fungal species, perhaps best illustrated in three species in the same genus (rather than two species in different families as here) by Wu et al. 2018 mBio doi.org/10.1128/mBio.01130-18.  This paper would be well worth considering in the Discussion.

Author response: Thanks! The mentioned paper is excellent for understanding regulatory role of the other CDP activator WetA among different Aspergillus species but not directly linked to our topic.

 (3) It would be good to expand on the reason no AbaA complementation strains could be generated.

Author response: We have characterized several hundreds of insect-pathogenic fungal genes in the past 10 years and occasionally failed to generate complementation mutants of four or five genes with unknown reasons. In the present study, we tried all possible strategies in many attempts to complement abaA into its disruption mutant during a period of 15 months but always failed with no reason to be understood. Therefore, we had to use three disruption mutants of abaA as replicates in the present study as we did occasionally in the previous studies.

 (4) There are numerous typographic or other edits that would improve the manuscript, some of which are listed below.

General ‘UDAP’: is this a standard abbreviation in the field?  It becomes cumbersome on lines 55-57 or 66 where the ‘A’ duplicates ‘activates’.

Author response: The upstream developmental activation pathway is often defined as UDA pathway but UDAP in this manuscript. For example, FluG is often mentioned as 'UDA activator' in literature. Either UDA or UDAP activator may not cause any confusion once it is defined in a paper. Anyway, the sentence in the mentioned lines has been rewritten as suggested.

Lines 61-62: could be ‘increasingly apparent that the principles elucidated in A. nidulans are applicable across the Pezizomycotina’.  Note this is a bit strange given the research performed on other models, like Neurospora crassa.  Worth considering other wording.

Author response: Whether the principles elucidated in A. nidulans are applicable to Pezizomycotina remains debatable, as shown in cited papers.

Line 67: ‘bassiana which is a main source’.

Author response: revised as suggested.

Line 76: ‘but are required for the’.

Author response: revised as suggested.

Line 78: ‘sensu lado’? likely delete.

Author response: It has been changed to 'complex'.

Line 97: probably should write as ‘To reveal whether the transcription of three…’ then ‘are sequentially regulated’.

Author response: revised as suggested.

Line 111: may need italics for ‘Ptef1’.

Author response: Yes. It has been changed to Ptef1.

Line 121: spelling ‘Sabouraud’.

Author response: it has been corrected.

Line 167: add space ‘(Ds) and’.

Author response: revised as suggested.

Line 196: delete ‘respectively’.

Author response: revised as suggested.

Line 246: could be ‘The brlA and abaA genes expressed in the wt strain showed differential time’.

Author response: revised as suggested.

Figure 2: the ‘A’ may need to be moved up to avoid overlap with the photograph.

Author response: revised as suggested.

Line 479: spelling ‘transcriptome’.

Author response: corrected as suggested.

Lines 480 and 482: clear text by deleting ‘extraordinary’ and ‘vital’.

Author response: deleted as suggested.

Line 489: ‘as they do in’.

Author response: revised as suggested.

Author response: corrected as suggested.

Table S1 text: delete extra .  ‘plasmid..’

Author response: done.

Reviewer 4 Report (New Reviewer)

The authors have characterized two developmental regulators of Metarhizium robertsii. One of them is the best BLAST hit for Aspergillus nidulans BrlA. The second one is the homolog of AbaA. Both proteins are key developmental regulators in A. nidulans, and BrlA is also known to be a master regulator, a key and essential element in the genetic/molecular pathways controlling asexual development in this model eurotiomycete. The authors characterize the expression levels of these two genes, subcellular localization of both proteins and the phenotype of the single-null mutants of these two genes under an array of stress conditions. They also quantify asexual spore production and changes in virulence of the two single-null mutants.

In my opinion, the work is of interest for the readers of Journal of Fungi. However, the quality of the language must be improved. As I use to do, I attach a pdf copy of the manuscript with my comments and corrections. I hope they will help, but the authors should carry out a comprehensive review of the text.

Additional issues are:

-The fact that the expression of wetA is increased before that of the two genes characterized here not necessarily means that wetA is not controlled (also) by AbaA and/or BrlA in M. robertsii. WetA could have acquired additional functions but still be controlled by these two regulators regarding the control of asexual development. I recommend rewriting.

-It is true that a BLAST of A. nidulans BrlA in the M. robertsii database gives as the first hit the hypothetic gene characterized by the authors. It is true, too, that a reverse BLAST of the M. robertsii protein against A. nidulans database gives BrlA as the first hit. However, the score, E and coverage values are very low, and limited only to some amino acids of the C2H2 domain. There are several C2H2-type transcription factors that share or belong to a common subfamily of C2H2 TFs (SteA, MsnA, BrlA, CreA… in A. nidulans). My question is: are these low score, E and coverage values enough to say that protein KHO11655 is the homolog of BrlA?

-I would like to see a deeper discussion on RNA-seq data, which points to a misregulation of secondary metabolism. The corresponding figure mentions, for example, austinol and dehydroaustinol, the latter being related to FluG activity and the induction of conidiation in A. nidulans.

-The discussion should be rewritten, since as it is, it is a summary of the results, and not a discussion of the implications and importance of the results obtained in this work.

Taking everything into consideration, my recommendation is major review

Author Response

Please see attached a file for our response to your review.

Round 2

Reviewer 4 Report (New Reviewer)

The authors have replied to my comments on the previous version of the manuscript. Regarding wetA, I insist on the possibility that there could be a sequential activation and that this is not incompatible with an earlier activation of wetA in the context of the development of new functions for WetA in M. robertsii. The authors have rewritten the text but the general idea is basically the same as in the previous version, and in my opinion somewhat biased.

Regarding M. robertsii brlA homolog, I recommend the authors to include: a) in Figure S2 the reading frame and the amino acid each codon encodes; and b) in the main text, E, score and coverage values for nucleotide and protein BLASTs, for M. robertsii brlA/BrlA and abaA/AbaA. That will give the reader quantitative data on the conservation level.

Overall, my recommendation is minor review.

Author Response

Thank you very much for reviewing our manuscript and your thoughtful comments. 

Comments and Suggestions for Authors

The authors have replied to my comments on the previous version of the manuscript. Regarding wetA, I insist on the possibility that there could be a sequential activation and that this is not incompatible with an earlier activation of wetA in the context of the development of new functions for WetA in M. robertsii. The authors have rewritten the text but the general idea is basically the same as in the previous version, and in my opinion somewhat biased.

Regarding M. robertsii brlA homolog, I recommend the authors to include: a) in Figure S2 the reading frame and the amino acid each codon encodes; and b) in the main text, E, score and coverage values for nucleotide and protein BLASTs, for M. robertsii brlA/BrlA and abaA/AbaA. That will give the reader quantitative data on the conservation level.

Overall, my recommendation is minor review.

Author response: Thank you very much for reviewing our manuscript. In Figure S2, the open reading frame starts from ATG and terminates at TGA. This is clear for the readers. Notably, the DNA sequence of BrlA has two ATG codes in A. nidulans (in th lines marked with -321 and 105e respectively) but only one ATG code in M. robertsii ( in line marked with 97).

The BLASTp information has been added to the first paragraph of Results as suggested, including the values of E, total score, coverage and sequence identity.

This manuscript is a resubmission of an earlier submission. The following is a list of the peer review reports and author responses from that submission.

Round 1

Reviewer 1 Report

In this study Zhang et al. identified the orthologs of Aspergillus brlA and abaA in Metarhizium robertsii, the major regulators of the asexual reproduction. The authors examined the transcription of that genes in different developmental stages and characterized the knock-out mutants, including conidiation and blastospore production, virulence, stress tolerance and transcriptomic profiles.

The manuscript is well-written, the figures are informative and while M. robertsii is used as a biocontrol agent against agricultural pests, deeper insight into the regulation of asexual reproduction is of great interest.

Major points:

Figure 1: subcellular localization of brlA and abaA was investigated with GFP fusion proteins. The quality of images (resolution) needs to be increased. On the other hand, on LSCM images fluorescence signal exist in the complete hyphae. Presumably, the proteins are localized in the nuclei thus the green color resulted by autofluorescence, which needs to be reduced. Moreover, colocalization on Figure1/4 is resulted by the intensive red signal of DAPI and not the GFP signal. The LSCM parameters (e.g., channels) must be also included in the Mat Methods.

Minor points/questions:

-A regulatory network figure would be useful in the part Introduction including the major regulators of conidiation in Aspergillus (including CDP, UDAP and downstream genes).

-In Aspergillus nidulans brlA consist of overlapping transcription units (brlAalfa, brlAbeta and a uORF located in the 5`UTR). Similar alternative transcripts of brlA also exist in Metarhizium? (if yes than would be nice to be described in section 3.1.)

-When activation of CDP genes was examined L:D 12:12 photoperiod incubation condition was used. RNA was isolated during the light or the dark period?

-It is not clear from the text why complementation of abaA in the deletion mutant was unsuccessful. Short explanation would be required here.

-Text needs to be revised; few typos exist.